# Phospholipids alter activity and stability of mitochondrial membrane-bound ubiquitin ligase MARCH5

Lisa Merklinger[1], Johannes Bauer[2], Per A Pedersen[3], Rune Busk Damgaard[1], J Preben Morth[1]

Mitochondrial homeostasis is tightly controlled by ubiquitination. The mitochondrial integral membrane ubiquitin ligase MARCH5 is a crucial regulator of mitochondrial membrane fission, fusion, and disposal through mitophagy. In addition, the lipid composition of mitochondrial membranes can determine mitochondrial dynamics and organelle turnover. However, how lipids influence the ubiquitination processes that control mitochondrial homeostasis remains unknown. Here, we show that lipids common to the mitochondrial membranes interact with MARCH5 and affect its activity and stability depending on the lipid composition in vitro. As the only one of the tested lipids, cardiolipin binding to purified MARCH5 induces a significant decrease in thermal stability, whereas stabilisation increases the strongest in the presence of phosphatidic acid. Furthermore, we observe that the addition of lipids to purified MARCH5 alters the ubiquitination pattern. Specifically, cardiolipin enhances auto-ubiquitination of MARCH5. Our work shows that lipids can directly affect the activity of ubiquitin ligases and suggests that the lipid composition in mitochondrial membranes could control ubiquitination-dependent mechanisms that regulate the dynamics and turnover of mitochondria.

## Introduction

The eukaryote ubiquitin (Ub) system of eukaryotes is highly complex and regulates virtually all aspects of cell biology and homeostasis through structurally and functionally diverse Ub modifications (Komander & Rape, 2012). Ub modification of a substrate protein is performed by an ATP-dependent three-enzyme cascade comprising the Ub-activating (E1) enzymes, the Ub-conjugating (E2) enzymes, and the Ub ligases (E3) (Kerscher et al, 2006). These enzymes catalyse the attachment of Ub via its carboxyl terminus to usually a substrate lysine, leading to mono-ubiquitination of the substrate. Additional ubiquitination of a Ub conjugated to a substrate leads to poly-ubiquitination giving rise to eight possible poly-Ub chain linkages through either the amino terminus (M1-linked chains) or any of the seven lysines (K) residues of Ub (K6-, K11-, K27-, K29-, K33-, K48-, or K63-linked chains) (Hochstrasser, 2006; Deol et al, 2019), which define the fate of the modified protein (Kliza & Husnjak, 2020). Over the past years, it has become clear that Ub-dependent regulation plays a crucial role in mitochondrial homeostasis by fine-tuning mitochondrial dynamics and removal of damaged mitochondria by mitophagy (Harper et al, 2018). Furthermore, ubiquitin-dependent quality control guides mitochondrial import and proteasomal degradation of mitochondrial proteins.

The Ub ligase MARCH5 is a critical regulator in mitochondrial quality control and homeostasis. It is one of only two known membrane-bound E3s located in the outer mitochondrial membrane (OMM). MARCH5 belongs to the membrane-associated RING-CH-type finger (MARCH) family of Ub ligases and harbours four predicted transmembrane (Yonashiro et al, 2006; Karbowski et al, 2007). It contributes to mitochondrial homeostasis by regulating protein import, mitochondrial dynamics and morphology, targeting proteins of the fusion and fission machinery (Nakamura et al, 2006; Yonashiro et al, 2006; Park et al, 2010; Phu et al, 2020). Mitochondrial fusion is regulated by MARCH5-dependent ubiquitination of Mitofusins, a dynamin-like class of protein engaged explicitly in the fusion event (Park et al, 2014; Kim et al, 2015). Mitochondrial fission is controlled by the degradation of fission factors like the mitochondrial dynamic protein of 49 kD (MiD49) linked to MARCH5-mediated ubiquitination (Xu et al, 2016; Cherok et al, 2017; Nagashima et al, 2019). In addition, MARCH5 controls mitochondria-ER contact sites by ubiquitination of Mitofusin-2 and thus facilitates lipid transfer (Sugiura et al, 2013; Nagashima et al, 2019). Furthermore, recent studies showed that MARCH5-dependent recruitment and regulation of Parkin, a cytosolic Ub ligase controlling mitophagy, adjusts Parkin dependent mitochondrial turnover (Koyano et al, 2019; Shiiba et al, 2021). However, it remains unclear how MARCH5 activity is regulated and how its finely tuned ubiquitination of specific substrates in different mitochondrial conditions is controlled.

Phospholipids have a crucial signalling function and regulate membrane proteins through lipid–protein interactions that control protein activity, folding, stability or localisation (Corradi et al, 2019). Mounting evidence shows that phospholipids regulate mitochondrial homeostasis by interaction with mitochondrial proteins and recruitment of targets from other organelles and the cytosol (Dudek, 2017). Recent studies revealed the impact of phosphatidic

[1]Department of Biotechnology and Biomedicine, Technical University of Denmark, Lyngby, Denmark   [2]Centre for Molecular Medicine Norway (NCMM), Nordic EMBL Partnership University of Oslo, Oslo, Norway   [3]Department of Biology, University Copenhagen, August Krogh Bygningen, Copenhagen, Denmark

Correspondence: premo@dtu.dk

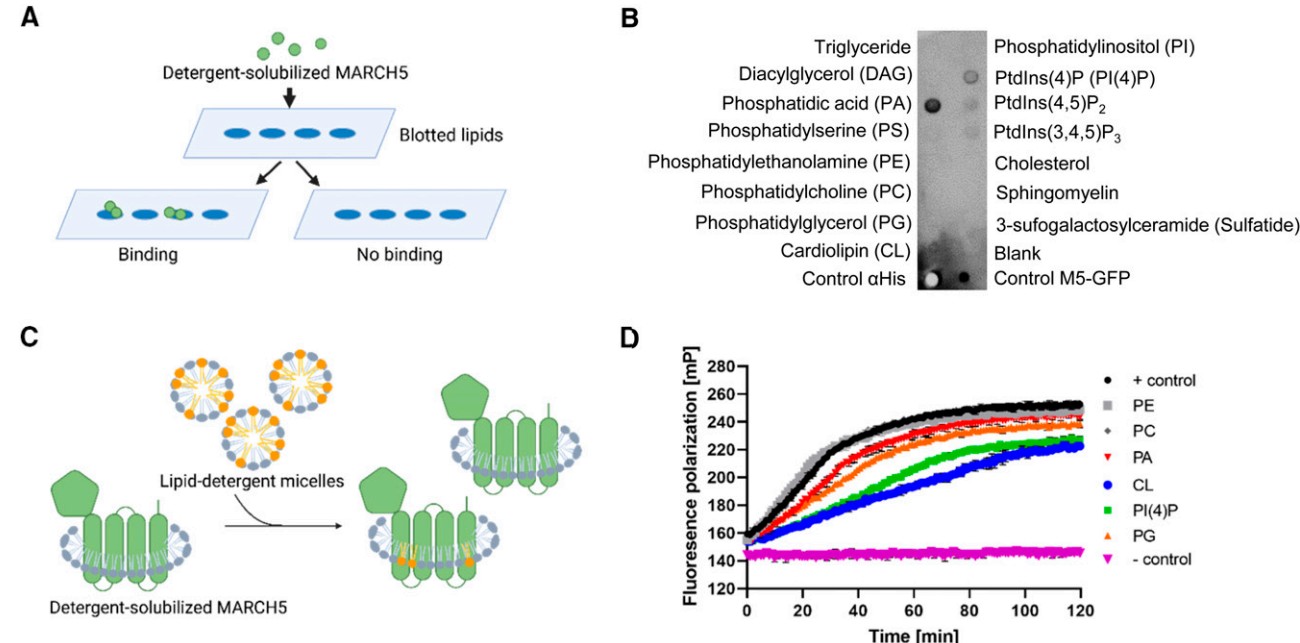

**Figure 1. Different lipid classes interact specifically with MARCH5 and alter ubiquitination pattern in vitro.**
**(A)** Schematic overview of a lipid-binding assay. **(B)** Purified MARCH5-GFP-His$_{10}$ in the detergent LMNG was incubated with a lipid membrane strip (Echelon) and MARCH5-lipid interactions were subsequently detected using an anti-His antibody. As controls, anti-His antibody and M5-GFP-His$_{10}$ was spotted on the membrane before the assay. The experiment was performed in three independent experiments. **(C)** Illustration of preparation of MARCH5 lipid samples. The sample is a mix of detergent solubilised-solubilized MARCH5 only, and MARCH5 interacting with lipids. **(D)** Fluorescence polarisation ubiquitination assay of MARCH5 in the presence of six different lipid classes (PE, PG, PI(4)P, PC, PA, or CL), tracking F-Ub over time. + control curve represents MARCH5 in the absence of lipids. – control included all reaction components except ATP. Data curves are one representative experiment of three independent experiments performed in four replicates shown as values ± SD.

acid (PA) and cardiolipin (CL) on mitochondrial dynamics (Frohman, 2015; Kameoka et al, 2018). PA regulates mitochondrial fusion and ER–mitochondrial contact sites of the OMM (Choi et al, 2006; Huang et al, 2011; Baba et al, 2014) and deactivates the mitochondrial fission protein dynamin-related protein 1 (Drp1) (Adachi et al, 2016, 2018). In contrast, CL stimulates mitochondrial fission by activation of Drp1 (Bustillo-Zabalbeitia et al, 2014; Mahajan et al, 2021) and Drp1-CL interactions induce CL-cluster formation, which could form hotspots for mitochondrial division (Stepanyants et al, 2015). Furthermore, CL has an essential role in inducing mitophagy upon oxidative stress, and exposure of its oxidised form regulates apoptosis (Chu et al, 2013, 2014; Praharaj et al, 2019).

Here, we show that different lipid classes can affect the activity and stability of an Ub ligase, namely MARCH5. We investigated how the activity of the membrane-associated MARCH5 is affected by specific lipids of the mitochondrial membranes. Our findings reveal that lipids, particularly CL, can regulate both the ubiquitin-conjugating activity and stability of MARCH5. This expands the repertoire of mechanisms for regulating ubiquitination processes and substantially increases our knowledge of the regulation of MARCH5-mediated ubiquitination.

## Results

### MARCH5 interacts with PA, CL, and PI(4)P

MARCH5 is a membrane protein, which interacts with phospholipids in the OMM over the hydrophobic and hydrophilic patches of its transmembrane regions (Contreras et al, 2011). Most lipid-membrane protein interactions are transient and likely facilitated by the abundant bulk lipids. Lipid classes binding preferably to the protein and are forming a lipid shell are referred to as annular lipids, whereas lipids binding to a specific site of the protein are referred to as non-annular lipids (Contreras et al, 2011). To analyse the binding of lipids to MARCH5, we investigated the lipid interaction of purified MARCH5 with major mitochondrial lipids using a lipid-binding assay (Fig 1A). MARCH5-GFP-His$_{10}$ was incubated with a polyvinylidene difluoride (PVDF) membrane spotted with different lipids, and MARCH5-lipid interactions were visualised with an anti-His antibody. The lipid-binding assay revealed that MARCH5 binds to PA, CL and PI(4)P (Fig 1B). Different detergents and the GFP-His$_{10}$-tag did not have any effect on the detected MARCH5-lipid interaction (Fig S1A and B).

The binding of PA and CL to MARCH5 raised the question of whether MARCH5 has specific binding sites. The MARCH5 sequence is highly conserved over all higher eukaryotic species (Bauer et al, 2017). Therefore, we looked at identified PA or CL binding proteins and their binding site. Analysed CL and PA binding proteins often have stretches of positively charged residue (Lys, Arg, and His) in their lipid-binding site (Musatov & Sedlák, 2017; Tanguy et al, 2018). Two recently identified potential PA binding motifs were described as (K/R/H) (K/R/H) and (K/R/H)X$_{1-3}$(K/R/H), whereas X can be any residue (Zhukovsky et al, 2019). We identified three similar motifs in the predicted transmembrane region of MARCH5, Lys249-Lys253, Lys253-Lys257, and Lys157-Lys160 (Fig S2). These binding sites overlap with the features of identified CL-binding sites, where

typically positive charged amino acids, Arg and Lys bind to the head group of CL, whereas the hydrophilic amino acids Leu, Ile, and Val are overrepresented in the binding of the acyl chains. However, CL-binding sites tend to have higher flexibility, indicated by a higher content of Gly (Planas-Iglesias et al, 2015).

## MARCH5-mediated ubiquitination reflects the phospholipid environment

Approximately 80% of the mitochondrial membranes consist of similar amounts of the bulk lipids phosphatidylcholine (PC) and phosphatidylethanolamine (PE). The remaining 20% are less abundant phospholipids, including phosphatidylinositol (PI), phosphatidylserine (PS), phosphatidic acid (PA), phosphatidylglycerol (PG), and cardiolipin (CL), a lipid specific of the inner mitochondrial membrane (IMM) (Horvath & Daum, 2013).

To examine if and how the lipid composition affects the activity of MARCH5, we performed an in vitro ubiquitination assay with purified and detergent-solubilised MARCH5 (Fig S3A and B) in the presence of lipids from six different lipid classes (PA, PC, PG, PE, CL, and PI(4)P) common to the mitochondrial membranes, tracking fluorescent labelled-ubiquitin over time using fluorescence polarisation (FP). The lipids used derived from natural sources and vary in the fatty acid chain length and saturation. Therefore, we can only conclude about the lipid class, not the lipid class species.

We incubated detergent-solubilised human MARCH5 with different lipid-detergent micelles (Fig 1C) in a 1:10 M ratio of protein to lipid before the start of the ubiquitination assay. We used a promiscuous E2 enzyme (UbcH5b) for the assay, which was previously shown to catalyse MARCH5-mediated ubiquitination (Phu et al, 2020).

We observed an increase in FP signal over time in the presence of the tested lipid samples and the + control (MARCH5 in the absence of lipid) (Fig 1D). A FP signal increase indicates higher molecular weight (Mw) species linked to the formation of ubiquitin chains (Franklin & Pruneda, 2019). Therefore, MARCH5 was active in the absence and presence of all tested lipids. This indicated that specific lipids are not required for the activity of MARCH5. However, we observed significant changes in FP signal over time compared with the + control when adding minor lipids (CL, PI(4)P, PA or PG) to MARCH5. CL and PI(4)P induced the most significant decrease in the final FP signal compared to the + control after 2 h. This relates to shorter ubiquitin chains because of smaller formed Mw species. MARCH5 in the presence of PA or PG showed a weaker decrease in the final FP signal compared with the + control. The FP ubiquitination curve of MARCH5 in the presence of PE or PC resembled the curve of MARCH5 in the absence of lipids (+ control) (Fig 1D), suggesting that there is no significant change in MARCH5 activity in the presence of a bulk lipid. Together, these results show clear lipid dependent MARCH5-mediated ubiquitination and emphasise the importance of further investigating MARCH5-lipid interaction and the effect on its activity.

## MARCH5 shows different levels of auto-ubiquitination in the presence of specific lipid classes

To investigate further the effect of the tested lipids on the activity of MARCH5, we performed a ubiquitination assay using SDS–PAGE and

Western blotting for gel-based analysis to obtain more detailed information on MARCH5-mediated ubiquitination. In addition, we quantified the intensity of the MARCH5 and mono-ubiquitin bands on the SDS–PAGE gel appearing at 25 and 10 kD, respectively, to monitor the level of ubiquitination (Fig 2C and D).

MARCH5 showed activity in the absence (+ control) and in the presence of all tested lipids, indicated by the depletion of mono-ubiquitin over time in the Coomassie-stained SDS–PAGE gels and the characteristic high Mw ubiquitination smear emerging after 15 min (Fig 2A and C). The complementary Western blot revealed that higher Mw ubiquitin species were formed after 15 min (Fig 2B). In addition, treatment of the ubiquitination reaction by a deubiquitinase reversed the poly-ubiquitin chain reaction, shown by a reappearance of the mono-ubiquitin band at 10 kD and reduction of the ubiquitin smear in the SDS–PAGE gel and Western blot, thereby confirming the formation of poly-ubiquitin chains (Fig S4A and B).

Interestingly, MARCH5 showed an increased level of auto-ubiquitination (ubiquitination of MARCH5 itself) in the presence of four of the tested lipids, which is shown by a decrease in MARCH5 band on the Coomassie-stained SDS–PAGE gel over time. The highest level of MARCH5 auto-ubiquitination was observed in the presence of CL. The intensity of the MARCH5 band was reduced by 80–20% after the 1 h long experiment (Fig 2A and D), suggesting substantially enhanced auto-ubiquitination. A similar effect was observed in the presence of PI(4)P, as the intensity of the MARCH5 band showed a 60% reduction to 40% after the 1 h experiment (Fig 2A and D). This is supported by Western blotting against MARCH5 (Fig 2B), where we observed the formation of higher Mw species and a decrease in the main MARCH5 band in the presence of CL and PI(4)P. In contrast, MARCH5 only showed a reduction in band intensity of up to 20% as + control or presence of abundant PE or PC. Furthermore, MARCH5 showed a trend towards enhanced auto-ubiquitination in the presence of PA and PG.

To analyse the concentration dependency of the MARCH5-binding lipids PA and CL on the activity of MARCH5, we performed the activity assay at different protein to lipid ratios (1:1, 1:5, 1:10, and 1:50 M ratio). A ubiquitination assay of MARCH5 in the presence of increasing PC concentration, representing MARCH5 in the presence of bulk lipids, served as an additional control experiment.

The ubiquitination assay, including different lipid concentrations, revealed a dependency on auto-ubiquitination in the presence of CL (Fig 3). Increasing concentrations of CL enhanced auto-ubiquitination of MARCH5. This is supported by the reduction in MARCH5 band intensity (Fig 3A) and appearance of high Mw bands after 15 min, detected by anti-ubiquitin and anti-MARCH5 Western blotting (Fig 3B). Already at a 1:5 M ratio of protein to lipid, the intensity of the MARCH5 band is reduced to 20% after the 2 h of running the experiment. At a 1:50 M ratio of protein to lipid, the MARCH5 band at 25 kD diminished entirely after 30 min, and mono-ubiquitin turnover significantly decreased (Fig 3C and D). However, MARCH5 Western blots using anti-MARCH5-antibodies showed an intense smear of MARCH5 (Fig 2B). This might be due to the aggregation of unfolded MARCH5, supported by visual precipitation of MARCH5 in the presence of CL in the 1:50 M ratio of protein to lipid. To explore if MARCH5 in the presence of CL forms higher order oligomeric species, we performed crosslinking experiments of

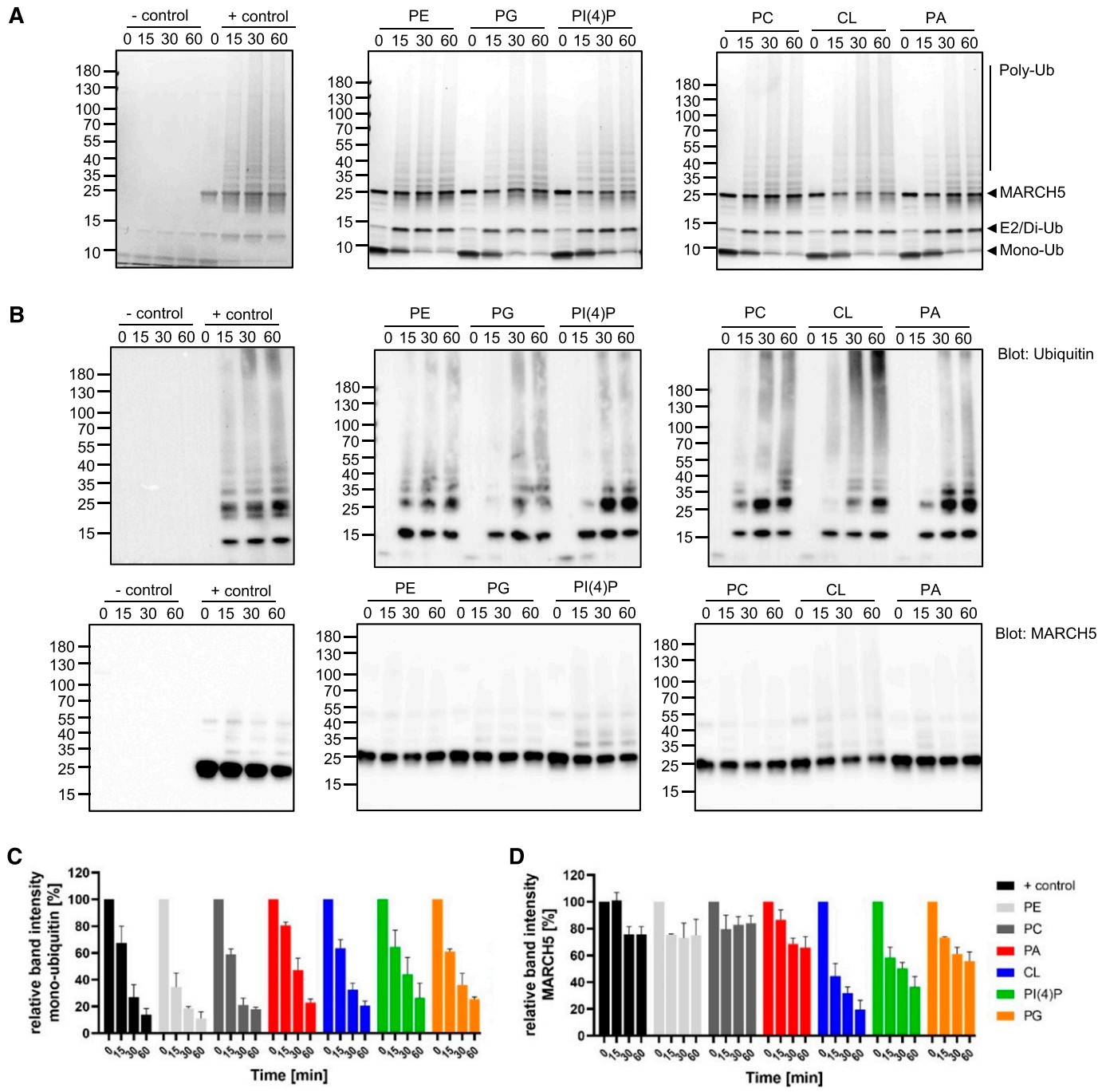

**Figure 2.  MARCH5-mediated ubiquitination is altered by the addition of lipids in vitro.**
**(A, B)** Coomassie-stained SDS–PAGE gel and (B) Western blot targeting ubiquitin (top row) and MARCH5 (bottom row) of ubiquitination assay of MARCH5 in the presence of lipids (PE, PG, PI(4)P, PC, PA, or CL) or in the absence of lipids (+ control) in a molar ratio of 1:10 protein to lipid. – control included all reaction components except MARCH5. The ubiquitination reactions were terminated at different time points (0, 15, 30, and 60 min). **(C, D)** Relative band intensities of MARCH5 at mono-ubiquitin at 10 kD (C) and MARCH5 at 25 kD (D) of Coomassie-stained SDS–PAGE gels. Time 0 min of each reaction solution represents 100% and reduction of either MARCH5 or mono-ubiquitin is represented relative to the respective time 0. Data information: In (A, B), experiments were performed as three independent experiments and (C, D) data are represented as mean values ± SEM of these three independent experiments.
Source data are available for this figure.

MARCH5 in the presence of CL in a 1:10 M ratio protein to lipid. We observed a trend towards increased dimerisation when compared with the control (Fig S4C).

In contrast, incubating MARCH5 in the presence of increasing PA concentration showed again an increasing trend towards increased auto-ubiquitination, especially in the 1:50 M ratio of protein to lipid

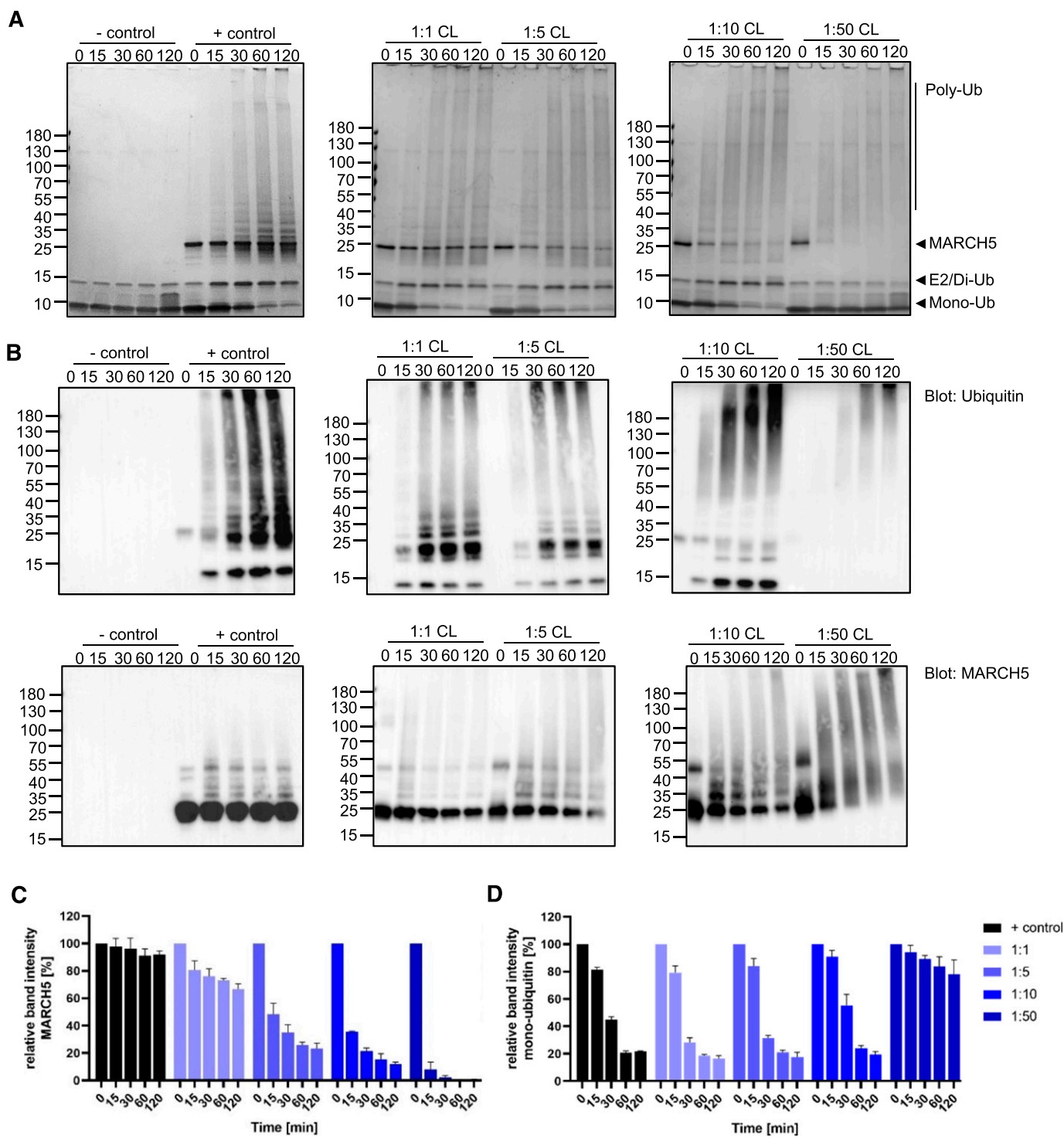

**Figure 3. Cardiolipin induces MARCH5 auto-ubiquitination.**
**(A, B)** Coomassie-stained SDS–PAGE and (B) Western blot visualised by anti-ubiquitin antibodies (top row) or anti-MARCH5 antibodies (bottom row) of samples taken from the MARCH5 ubiquitination assay in presence of CL at 1:1, 1:5, 1:10, and 1:50 M ratio of protein to lipid or in the absence of lipids (+ control). – control included all reaction components except MARCH5. The ubiquitination reaction was terminated at different time points (0, 15, 30, 60, and 120 min). **(C, D)** Relative band intensities of MARCH5 at 25 kD (C) and mono-ubiquitin at 10 kD (D) of Coomassie-stained of SDS–PAGE gels. Time 0 min of each reaction solution represents 100%, and reduction of either MARCH5 or mono-ubiquitin is represented relative to their respective band at time 0. Data information: In (A, B), experiments were performed as three independent experiments and (C, D) data were represented as mean values ± SEM of these three independent experiments.
Source data are available for this figure.

(Fig S5A–C and E). Here, MARCH5 band intensity is reduced to 40% after the 2 h long experiment (Fig S5C). When titrated with PC, MARCH5 did not show any significant differences in activity upon increased lipid concentration (Fig S5A, B, E, and F).

### MARCH5 stability is affected by the phospholipid environment

To address how the stability of MARCH5 is affected by the selected lipids, we used differential scanning fluorimetry (DSF) to measure thermal melting. Thermal melting was assessed by measuring the change in intrinsic fluorescence of mainly Trp at 330 and 350 nm with an increase in temperature, tracking the reconfiguration during denaturation. The polarity of the local microenvironment of MARCH5 is strongly influenced by the intrinsic fluorescence signal from mainly Trp (Real-Hohn et al, 2020; Cecchetti et al, 2021).

We initially compared temperature-induced melting profiles of MARCH5 in the presence of the lipids tested in the activity assay to the control sample (in the absence of lipid) (Fig 4 and Table 1). MARCH5, in the absence of lipids, had thermal stability of 52.9°C, which is defined by the transition point of the DSF curve and peak of its derivative. Significant, lipid specific changes in thermal stability of MARCH5 were observed in the presence of the PA, CL, PG and PI(4)P in a 1:10 M ratio of protein to lipid (Fig 4A). Surprisingly, CL more abundant in the IMM induced a significant 12.6°C decrease in thermal stability relative to the control, which is shown by a shift of the DSF curve towards lower temperature. The most significant increase in melting temperature was observed in the presence of PA with an increase in melting temperature of 7.3°C compared with the control, represented by a shift of the DSF curve towards higher temperature (Fig 4A and Table 1). PI(4)P and PG induced a 2.4°C and 3.4°C increase in the thermal stability of MARCH5, respectively, compared with the control. The most abundant lipids PC and PE did not significantly affect the melting temperature compared with detergent-solubilised MARCH5 and resemble the DSF and its derivation curve (Fig 4A).

Because CL and PA showed specific interaction and the strongest and interestingly opposing effects on MARCH5 stability, we examined the concentration dependency of MARCH5 towards these two lipids. To this end, we performed thermal stability measurements, using different protein to lipid ratios (1:1, 1:5, 1:10, and 1:50 M ratio) (Fig 4B–D), as previously described. Again, titration of PC, an abundant lipid in the OMM, served as an additional control experiment.

Lipid titration of PA and CL revealed a concentration dependency of MARCH5 stability (Table 2 and Fig 4B and C), supporting the hypothesis of specific MARCH5-lipid interactions. MARCH5 in the presence of CL in a 1:5 M ratio of protein to lipid already showed a significant thermal stability reduction of 10.3°C compared with MARCH5 in the absence of lipids. A further increased CL concentration led to a decrease in the thermal stability of MARCH5. At the highest tested molar ratio of protein to lipid (1:50), MARCH5 even showed a 23.2°C reduction in melting temperature compared with the control (Fig 4B). In contrast, PA in presence in a 1:5 M ratio of protein to lipid increased MARCH5 stability significantly by 6.1°C, whereas a 1:10 M ratio resulted in the maximum stabilisation of 8.0°C compared with MARCH5 in the absence of lipids (Fig 4C). A higher PA concentration did not further increase the thermal

stability of MARCH5. Titration of PC did not show a significant change in thermal stability of MARCH5 (Fig 4D and Table 2).

## Discussion

MARCH5 has a vital role in mitochondrial homeostasis and has been shown to regulate mitochondrial morphology by ubiquitination of fission and fusion proteins (Yonashiro et al, 2006; Park et al, 2014; Kim et al, 2015; Xu et al, 2016; Cherok et al, 2017; Phu et al, 2020). However, regulation of MARCH5 facilitating customised ubiquitination of a specific substrate to generate the desired biological outcome has remained largely unclear. Regulation of Ub ligases often occurs at the level of post-translational modification, small molecule modulators and metabolites (Deshaies & Joazeiro, 2009; Metzger et al, 2014). MARCH5 is one of only two integral membrane Ub ligases in the OMM, restricted to the two-dimensional lipid bilayer. The notion that the lipid environment highly controls membrane proteins was slow to be embraced but is now well established for multiple systems (Contreras et al, 2011; Martens et al, 2018; Patrick et al, 2018). However, how lipid environments or small modulators affect membrane-bound Ub ligase activity remains inconclusive. The recent study by Sharpe et al (2019) connected a specific lipid environment to a change in membrane-bound Ub ligase activity. They showed that the catalytic ability and stability of the membrane-bound Ub ligase MARCH6, another member of the MARCH family, is regulated by cholesterol (Sharpe et al, 2019). Furthermore, the lipid sphingosine-1-phosphate has been shown to be an activating cofactor of the Ub ligase TRAF2. TRAF2 is, however, not a membrane-associated Ub ligase (Alvarez et al, 2010).

In this study, we investigated the effect of the lipid environment on the activity and stability of the Ub ligase MARCH5. Through our in vitro activity, stability, and lipid binding experiments of MARCH5 in the presence of various lipid classes (CL, PA, PG, PI(4)P, PE, and PC), we got an insight into the possible regulatory roles of lipids in membrane-bound Ub ligase activity.

As expected, we did not observe any significant effects of the abundant lipids PE or PC on MARCH5 activity and thermal stability. This is reasonable if PC and PE are considered as bulk lipids, transiently interacting in a non-specific manner with the hydrophobic and hydrophilic patches of MARCH5. In contrast, the tested less abundant lipids CL, PA and PI(4)P bind MARCH5 and induced clear changes in MARCH5 activity and thermal stability. The binding of PI(4)P to MARCH5 increased its stability and induced a change in MARCH5 activity also shown by increased auto-ubiquitination. Despite no significant changes in MARCH5 activity in the presence of PA based on the SDS–PAGE gel and Western blot, we observed the most substantial stabilisation and the change in FP signal of MARCH5 in the presence of PA.

Furthermore, CL induced the most significant changes in stability and activity. These findings demonstrate that the lipid environment and specific lipids can regulate membrane-bound Ub ligase activity and stability in vitro. However, further investigation is necessary to unravel possible regulatory mechanisms of lipids towards membrane-bound E3 ligase activity and in vivo studies have to be performed to investigate the effect in a cellular context.

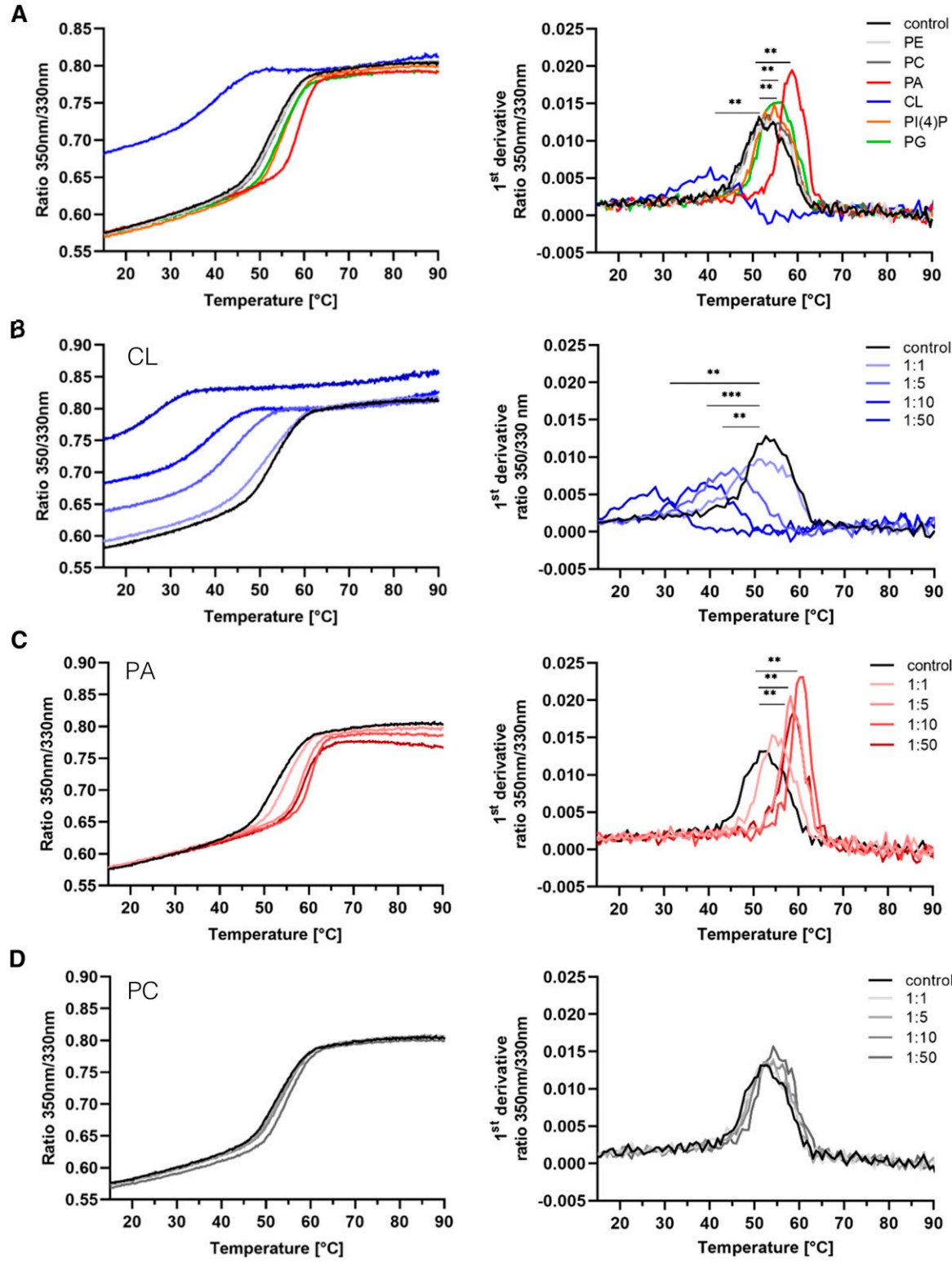

**Figure 4. MARCH5 reveals different thermal stability profiles depending on lipids.**
**(A, B, C, D)** Nano-differential scanning fluorimetry (nanoDSF) curves (left) and their first derivative (right) of purified MARCH5 in the presence and absence (control) of lipids. Samples were excited at 280 nm and the Trp emission at 330 and 350 nm was recorded with an increase in temperature. **(A)** Melting profile of MARCH5 in the presence of PE, PC, PA, CL, PG, or PI(4)P in a 1:10 M ratio of protein to lipid. **(B, C, D)** Lipid titration of MARCH5 in the presence of CL (B), PA (C), and PC (D) in different molar protein to lipid ratios (1:1, 1:5, 1:10, and 1:50). MARCH5, including the lipids, were incubated for 1 h at 4°C before measurements. Melting curves were obtained using the Prometheus Panta (Nanotemper) in a temperature range of 15°C–95°C and a temperature gradient of 1°C per minute. Experiments were performed three times, each in triplicates. Curves show one example for clarity. A paired $t$ test was applied. **$P < 0.01$, ***$P < 0.001$.

**Table 1. Melting temperatures of MARCH5 in presence of PC, PE, PI(4)P, PG, PA, or CL determined by nanoDSF.**

|  | Melting temperature (°C) |
|---|---|
| Control | 52.86 ± 0.58 |
| PC | 53.85 ± 0.27 |
| PE | 53.42 ± 0.47 |
| PI(4)P | 55.17 ± 0.57** |
| PG | 56.18 ± 0.55** |
| PA | 60.15 ± 0.74** |
| CL | 40.20 ± 0.54*** |

The melting temperatures of MARCH5 in the absence (control) or presence of different lipid classes in a 1:10 M ratio of protein to lipid were determined as the maxima of the first derivative of the 350 nm/330 nm ratio (shown in Fig 4). A paired *t* test was applied. **$P < 0.01$, ***$P < 0.001$. Data are shown as mean values ± SD of four independent experiments, each performed in triplicates.

**Table 2. Melting temperature of MARCH5 in presence of CL, PA, or PC determined by nanoDSF.**

|  | Melting temperature (°C) | | |
|---|---|---|---|
| Molar ratio | CL | PA | PC |
| Control | 53.20 ± 0.60 | 52.91 ± 0.65 | 52.96 ± 0.71 |
| 1:1 | 52.08 ± 0.15 | 55.24 ± 0.21 | 52.53 ± 0.28 |
| 1:5 | 42.87 ± 0.96** | 58.98 ± 0.32** | 53.17 ± 0.08 |
| 1:10 | 38.95 ± 1.43*** | 60.86 ± 0.26** | 54.01 ± 0.16 |
| 1:50 | 29.98 ± 3.67** | 59.57 ± 1.11** | 55.53 ± 0.48 |

The melting temperature of MARCH5 in the absence (control) or presence of CL, PA, and PC in different protein to lipid molar ratios was determined as the maxima of the first derivative of the 350 nm/330 nm ratio (shown in Fig 4). A paired *t* test was applied. $P < 0.01$**, ***$P < 0.001$. Data are shown as mean values ± SD of three independent experiments, each performed in triplicates.

Intriguingly, as previously mentioned, MARCH5 showed the most elaborate changes in stability and activity in the presence of CL, even though CL is a specific lipid of the IMM. The increased auto-ubiquitination of MARCH5 observed in the presence of CL is likely connected to the decreased thermal stability in the presence of CL. Binding of CL to MARCH5 might induce structural changes in MARCH5, leading to auto-ubiquitination. This indicates that the presence of CL actively induces ubiquitination of MARCH5 and therefore alters MARCH5-mediated ubiquitination.

This proposed regulatory mechanism gives interesting new aspects to MARCH5 regulation in the mitochondria, where CL is only present in the OMM in trace amounts in the steady-state cell (Horvath & Daum, 2013). However, under mitochondrial stress, CL can be externalised to the OMM, serving as a signal for mitophagy or mitochondrial fission (Chu et al, 2013, 2014). The fission of damaged and dysfunctional mitochondrial segments before mitophagy is essential for the engulfment by phagosomes (Twig et al, 2008; Dudek, 2017). Recent studies demonstrated that the specific interaction of the mitochondrial fission protein Drp1 with externalised CL is important for Drp1's activity and recruitment to damaged mitochondria, allowing CL-induced fission (Bustillo-Zabalbeitia et al, 2014; Stepanyants et al, 2015; Mahajan et al, 2021). Interestingly, MARCH5-depleted cells showed

increased mitochondrial fragmentation because the accumulation of mitochondrial fission factors like Drp1 and MiD49 (Xu et al, 2016; Nagashima et al, 2019). Furthermore, Drp1-mediated mitochondrial fission is regulated by MARCH5-mediated degradation of the fission protein MiD49 (Cherok et al, 2017). Therefore, we speculate that under mitochondrial stress and externalisation of CL to the OMM, MARCH5 binds to CL, which might lead to ubiquitination and, therefore alternation of MARCH5 activity. This would allow the recruitment of mitochondrial fission machinery such as Drp1 and MiD49 to the damaged mitochondria site. Therefore, we suggest CL as a regulator of MARCH5 upon externalisation to the OMM, especially in CL-fission (Fig 5).

To summarise, we showed that lipids could alter membrane-bound Ub ligases on the mitochondrial ligase MARCH5 in vitro. We hypothesise that under different mitochondrial like fission and fusion, where lipid composition changes (Kameoka et al, 2018), MARCH5 activity is altered by these lipids, leading to finely tuned MARCH5-mediated ubiquitination in different mitochondrial processes.

In conclusion, we present the first study, where a variety of different lipid classes were tested to investigate the effect of lipids on the activity and stability of a membrane-bound Ub ligase. We elucidated a possible regulatory mechanism of CL on MARCH5 in the mitochondria. Furthermore, these results highlight the importance of the lipid environment in the ubiquitin system and open up further questions like substrate specificity or E2/E3 pairing upon a change in a lipid environment.

# Materials and Methods

All chemicals were purchased as grade BioUltra from Sigma-Aldrich if not stated otherwise. Lipids were purchased from Avanti Polar Lipids dissolved in chloroform.

### Generation of MARCH5

All cDNAs were purchased from GenScript, and codon-optimised for overexpression in yeast (*Saccharomyces cerevisiae*). For large-scale overexpression in yeast, the cDNA encoding human MARCH5 was inserted into the pEMBLyex4 plasmid (Cesareni & Murray, 1987) by homologous recombination in *S. cerevisiae* strain PAP1500 (α ura3-52 trp1::GAL10-GAL4 lys2-801 leu2_1 his3_200 pep4::HIS3 prb1_1.6R can1 GAL) (Pedersen et al, 1996) to encode MARCH5 followed by a TEV-cleavable yEGFP and a His$_{10}$-tag. DNA-sequencing verified the correct assembly of the expression construct. For homologous recombination, cDNAs for MARCH5 and yEGFP were amplified by PCR and co-transformed into *S. cerevisiae* PAP1500 with *Pst*I, *Hin*dIII, and *Bam*HI digested pEMBLyex4 vector following the protocol of Gietz and Schiestl (Gietz & Schiestl, 2007). The following primers were used:

March5_fwd: 5′-ACACAAATACACACACTAAATTACCGGATCAATTCTAA GATAATTATGCCAGATCAAGCATTGCAACAA-3′5′-ACACAAATACACACACT AAATTACCGGATCAATTCTAAGATAATTATGCCAGATCAAGCATTGCAA-CAA-3′; March5_rev:5′-AAATTGACTTTGAAAATACAAATTTTCGGCTTCTTCTTGT TCTGGGTAGTT-3′5′-AAATTGACTTTGAAAATACAAATTTTCGGCTTCTTCTT GTTCTGGGTAGTT-3′; GFPfwTEV:5′-GAAAATTTGTATTTTCAAAGTCAATT-TATGTCTAAAGGTGAAGAATTATTCACT-3′.

**Life Science Alliance**

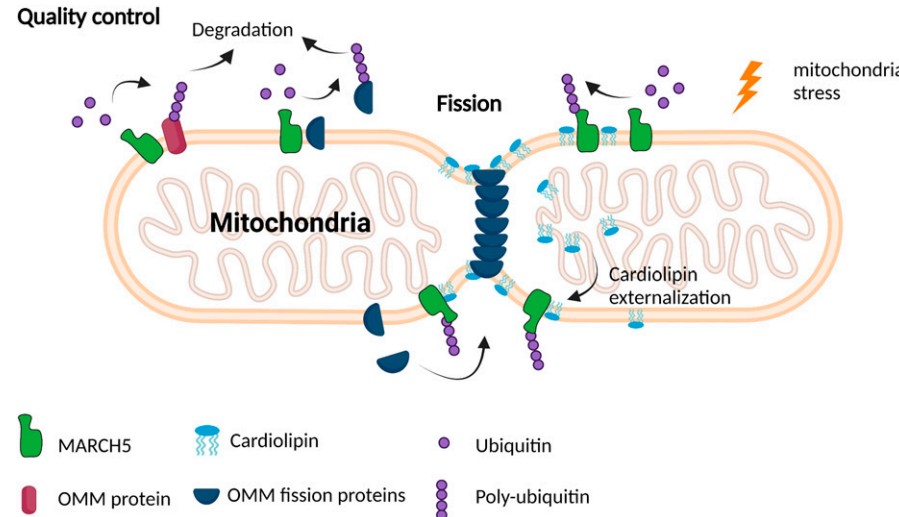

**Figure 5. Schematic model of CL regulated MARCH5-mediated ubiquitination in mitochondrial fission.** MARCH5 contributes to mitochondrial quality control by ubiquitination of misfolded proteins in the outer mitochondrial membrane. In steady-state mitochondria, MARCH5 negatively regulates mitochondrial fission proteins by ubiquitination, leading to degradation. Under mitochondrial stress, CL can be externalised, which might induce ubiquitination of MARCH5. This would allow proteins of the fission machinery to target the fission site and the fission of the damaged part of the mitochondria could take place.

GFPrecrv:5'-CTTCAATGCTATCATTTCCTTTGATATTGGATCATTCAATGG TGATGGTGATGGTGATGGTGATGGTGTTTGTACAATTCATCCATACCA-3'.

## Expression and membrane extraction of MARCH5

Large-scale production of MARCH5 was carried out in *S. cerevisiae*. First, a single colony of transfected cells was selectively propagated at 30°C until saturation in 5 ml of minimal medium (0.5% ammonium sulfate, 2% glucose, salts [64 mM $KH_2PO_4$, 7 mM $K_2HPO_4$, 20 mM $MgSO_4$, 17 mM NaCl, 70 $\mu$M $CaCl_2$, 8 $\mu$M $H_3BO_3$, 0.6 $\mu$M KI, 2 $\mu$M $MnSO_4$, 7 $\mu$M $ZnSO_4$, 2 $\mu$M $CuSO_4$, 1 $\mu$M $FeCl_3$, and 1 $\mu$M $Na_2MoO_4$], vitamins [20 $\mu$g/l biotin, 2 mg/l pantothenic acid, 2 $\mu$g/l folic acid, 10 mg/l myo-inositol, 400 $\mu$g/l niacin, 200 $\mu$g/l 4 aminobenzoic acid, 400 $\mu$g/l pyridoxine, 200 $\mu$g/l riboflavin, and 400 $\mu$g/l thiamin], and 100 $\mu$g/l ampicillin) supplemented with L-leucine and L-lysine (both at 30 mg/l) followed by propagation into minimal medium supplemented only with L-lysine. 2 ml of this culture was then used to inoculate 250 ml minimal medium supplemented with L-lysine. For expression, 2.5 liters YPG medium (1% yeast extract, 2% peptone, and 3% glycerol) supplemented with 0.5% glucose, 70 $\mu$M $CaCl_2$, vitamins and ampicillin was first inoculated to $OD_{600}$ = 0.05 in a 5-L-baffle flask and cultivated for 24 h overnight at 30°C. At $OD_{600}$ = 2.0–4.0 the culture was then cooled down to 15°C and subsequently induced at $OD_{600}$ = 2.5–4.0 with 250 ml induction medium (YPG medium supplemented with 20% galactose). Cells were harvested 48 h after induction by centrifugation (4,000$g$, 15 min, 4°C). The resulting cell pellet (ca. 80 g cell wet weight from a 9-L culture) was resuspended in Milli-Q water, aliquot in 50 ml Falcons and centrifuged at 4,000$g$ for 10 min. The resulting cell pellet was stored at –80°C before membrane extraction. All following steps were performed a 4°C. Typically, 30 g cells were thawed and resuspended in resuspension buffer (50 mM Tris–HCl pH 7.7, 400 mM NaCl, 10% [vol/vol] glycerol, 1 mM EDTA, and 1 mM EGTA, freshly added 20 mM DTT, 5 mM PMSF, leupeptin [1 $\mu$g/ml], pepstatin A [1 $\mu$g/ml], and chymostatin [1 $\mu$g/ml] [L,P,C-protease inhibitor mix]) 1.2 times the volume of the cell pellet weight. Cells were lysed by bead beating. Therefore, resuspended cells were distributed equally into six 50 ml

Falcon tubes and then saturated with glass beads ($\phi$ = 500–750 $\mu$m, Acros Organics). Every tube was vortexed 5 × 1 min with at least 1 min incubation on ice between each cycle. Beads were then washed with 100 ml lysis buffer, and the pooled cell lysate was centrifuged twice (4,000$g$, 10 min, 4°C). The resulting supernatant was centrifuged (215,000$g$, 60 min, 4°C). The membrane pellet were homogenized in a ratio of 1 g per 10 ml solubilisation buffer (50 mM Tris–HCl pH 7.7, 400 mM NaCl, 10% [vol/vol] glycerol, 1 mM TCEP, 0.1 mM EDTA, 0.5 mM PMSF, and L,P,C protease inhibitor mix) using a 30-ml tissue grinder (Wheaton). The homogenized crude membranes were flash-frozen and stored at –80°C.

## Purification of MARCH5

All following steps were performed a 4°C. Crude membranes were thawed, supplemented with solubilisation buffer and finalised with 10% (wt/vol) n-Dodecyl $\beta$-D-maltoside ($\beta$-DDM; Anatrace) to a detergent concentration of 1% (wt/vol), so the final volume yielded 50 ml per g crude membranes. MARCH5 was solubilised for 1 h at 4°C, whereas stirring followed by centrifugation (22,000$g$, 40 min, 4°C). The supernatant was batch-incubated for 1 h with 1 ml of Nickel High-Performance resin (GE Healthcare) per 50 ml supernatant equilibrated in wash buffer 1 (50 mM Tris–HCl, pH 7.7, 100 mM NaCl, 10 mM imidazole, 5% (vol/vol) glycerol, 0.1 mM TCEP, and 0.01% (wt/vol) 2,2-didecylpropane-1,3-bis-$\beta$-D-maltopyranoside (LMNG, Anatrace)) in the presence of 10 mM imidazole. The resin supernatant mix was transferred to an empty PD-10 column resulting in a CV of 1 ml. Subsequently, the resin was washed with 20 column volumes (CV) of Wash buffer 1 followed by 20 CV of Wash buffer 2 (50 mM Tris–HCl pH 7.7, 400 mM NaCl, 90 mM imidazole, 5% [vol/vol] glycerol, 0.1 mM TCEP, and 0.01% [wt/vol] LMNG). Detergent was exchanged on the Nickel column to LMNG. The protein elution was carried out with three CV Elution buffer (50 mM Tris–HCl, pH 7.7, 100 mM NaCl, 500 mM imidazole, 5% [vol/vol] glycerol, 0.5 mM TCEP, and 0.001% [wt/vol] LMNG). GFP-tag was cleaved of MARCH5 using TEV-protease (homemade, 1:2-M ratio), while dialysing against two times 70 volumes of SEC buffer (50 mM Tris–HCl, pH 7.7, 100 mM NaCl, 5%

[vol/vol] glycerol, and 0.1 mM TCEP, 0.001% [wt/vol] LMNG) using a Snakeskin dialysis tubing (3.5 kD molecular weight cut off; Thermo Fisher Scientific), for MARCH5-GFP-His$_{10}$ tag no TEV protease was added during the buffer exchange. Buffer was exchanged after 1 h, and dialysis continued for another 18 h. The precipitation was removed by centrifugation (4,000$g$, 10 min, 4°C). To remove the cleaved GFP and TEV, the supernatant was incubated with 1 ml Nickel High Performance resin equilibrated in SEC buffer for 30 min. Flow-through and wash (2 CV with SEC buffer) containing cleaved MARCH5 was collected and concentrated using an Amicon Ultra 15 molecular weight cut-off 50 kD to around 1 mg/ml. After, analytical quality control by size exclusion chromatography using a Superose 6 increase 10/300 GL column (GE Healthcare) in SEC buffer, MARCH5 was flash-frozen in liquid nitrogen and stored at −80°C. For MARCH5 in DDM, the same purification procedure was used. However, the detergent was not exchanged and instead of LMNG, 0.01% (wt/vol) DDM in the buffers was used.

### Lipid-binding assay

For positive controls, 1 $\mu$l of anti-His-HRP (#7074; Cell Signalling) and 1 $\mu$l of MARCH5-GFP-His$_{10}$ in two concentrations (0.35 and 3.5 mg/ml) were pipetted on an empty spot on the lipid membrane strip (# P-6002; Echelon Bioscience). The dry membrane was blocked with 30 ml TBS + 3% (wt/vol) BSA overnight at 4°C, followed by incubation of 1 $\mu$g/ml MARCH5-GFP-His$_{10}$ in TBS + 3% (wt/vol) BSA and 0.01% (wt/vol) DDM or 0.001% (wt/vol) LMNG to ensure MARCH5 will not precipitate during incubation for 4 h at 4°C with soft agitation. The membrane was washed three times with TBS for 5–10 min, subsequently incubated with His-HRP antibody (1:2,000 dilution) in TBS + 3% (wt/vol) BSA for 1 h at room temperature. The previous wash step was repeated, followed by chemiluminescent development (Pierce ECL Western Blotting Substrate, #32132; Thermo Fischer Scientific) using Vilber Smart imaging Fusion Fx spectra or AEC staining Kit (#AEC101-KT; Sigma-Aldrich). For the lipid membrane strip using specific lipids, the 1 $\mu$l of lipid (PE (egg, chicken) (#840021), CL (heart, bovine) (#840012), PA (egg, chicken) (#840101), PC (egg, chicken) (#840051), PG (egg, chicken) (#841138), PI(4)P (brain, porcine) (#840045)) was spotted on a PVDF membrane and dried for 1 h. 1 $\mu$l of MARCH5 was spotted, serving as a positive control. The assay was performed as described above using 1 $\mu$g/ml MARCH5 in TBS + 3% (wt/vol) BSA and 0.001% (wt/vol) LMNG. However, for detection, the membrane was incubated with anti-MARCH5 (1:500, #ab185054; Abcam) and secondary antibody from anti-rabbit HRP (1:2,000, #7074; Cell Signalling) and detected by chemiluminescence using Biorad ChemiDoc Touch gel Imaging System.

### Lipid preparation

PE (egg, chicken) (#840021), CL (heart, bovine) (#840012), PA (egg, chicken) (#840101), PC (egg, chicken) (#840051), and PG (egg, chicken) (#841138) were purchased as stock solutions of 10 mg/ml. PI(4)P (brain, porcine) (#840045) was purchased as a 1 mg/ml stock solution. Notably, each lipid class included in this study consisted of a mixture of lipids varying in their chain length and saturation (e.g., CL mix of 18:1 and 18:2 fatty acid chains). The desired amounts of lipids dissolved in chloroform were dried under an argon stream.

The lipid films were resuspended in 0.2% (wt/vol) LMNG to a stock solution of 5 mM for a 1:10 M ratio of protein to lipid. For the titration curves of PA, PC, and CL, a separate lipid stock for each molar ratio (1:1, 1:5, 1:10, and 1:50) was prepared to a concentration that 5 $\mu$l of resuspended lipids to 35 $\mu$l MARCH5 sample resolve in the wanted molar ratio of protein to lipid (final volume of 40 $\mu$l). In the case of a used concentration of 32 $\mu$M MARCH5, a lipid stock solution of 0.19 mM for a 1:1 M ratio, 0.97 mM for a 1:5 M ratio, 1.94 mM for a 1:10 M ratio, and 9.68 mM for a 1:50 M ratio, was prepared. This allowed a constant detergent concentration in all molar ratios. To the control MARCH5 identical amount of detergent was added as for the lipid samples.

### In vitro ubiquitination assay for gel-based analysis

MARCH5 was incubated with lipids at a 1:10 M ratio on ice for 60 min (see lipid preparation section) before being added to the master solution. Therefore, 1.9 $\mu$l of 5 mM corresponding lipid stocks were incubated with MARCH5 (32 $\mu$M) to a final volume of 30 $\mu$l. MARCH5 in the absence of lipids (+ control) included the same amount of LMNG as in the samples with lipid, so detergent concentration remained constant throughout the different MARCH5 lipid samples and negative control containing no E3 ligase. A master solution resulting in final concentration of 40 mM Tris–HCl, pH 7.5, 25 mM NaCl, 10 mM MgCl$_2$, 0.6 mM DTT, 2 mM ATP, 50 nM human E1 (#E-304; Boston Biochem), 1 $\mu$M E2 (UbcH5b, #E2-622; Boston Biochem), and 23 $\mu$M Ub (#U6253; Sigma-Aldrich) was mixed and aliquoted. 2.5 $\mu$M of MARCH5 incubated with lipids was added to the respective master solution. The reaction volume was 60 $\mu$l per reaction mix. The final detergent and lipid concentration in the master solution was 1 CMC (critical micelle concentration) (0.001%), 25 $\mu$M, respectively. For the lipid titration ubiquitination assay, it was the same experimental setup, except the reaction volume was 80 $\mu$l per reaction mix and the preparation of the MARCH5 lipid samples. 5 $\mu$l of desired stock concentration of lipids (see lipid preparation) to achieve the wanted molar ratio of protein to lipid (1:1, 1:5, 1:10, and 1:50) was added to MARCH5 (32 $\mu$M), having a final volume of 40 $\mu$l and incubated 1 h on ice prior assay. The final detergent concentration was two CMC (0.0019%), and lipid concentration in the ubiquitination assay varied between 2.5 and 125 $\mu$M. Reactions were performed at 30°C for 0–60 min or 0–120 min, and for each time point, 14 $\mu$l reaction mix was taken out and stopped by the addition of 4 $\mu$l LSB. Samples were subjected to SDS–PAGE (8 $\mu$L sample loaded on Mini protean TGX gels 4–20%, #4561096; Bio-Rad), and proteins were visualised by Coomassie blue staining (InstantBlue Coomassie protein stain, #ab119211; Abcam) or immunoblotting (2.5 $\mu$l sample loaded on gel) using Bio-Rad ChemiDoc Touch gel Imaging System, anti-ubiquitin (1:2,000, #NB300-130; Novusbio) and anti-MARCH5 (1:500, #ab185054; Abcam) and secondary antibody from anti-rabbit HRP (1:2,000, #7074; Cell Signalling), anti-mouse HRP (1:4,000, #NA931; GE Healthcare). For relative quantification of the intensity of MARCH5 and mono-ubiquitin bands, the Image laboratory software (Bio-Rad) was used, and band intensity at time point 0 of the respective samples was defined as 100%. To note, because a substrate of MARCH5 was not available for the assay, we used the ability of many Ub ligases to facilitate ubiquitination even in the absence of their substrates, forming either free ubiquitin

chains, auto-ubiquitination or the E1 or E2 enzymes (Lorick et al, 1999).

### In vitro de-ubiquitination assay for gel-based analysis

Deubiquitinase USP2 (#E-504; Boston Biochem) was diluted to 2.5 $\mu$M in DUB buffer (150 mM NaCl, 50 mM Tris–HCl, pH 7.5, 20 mM DTT, and 1 mM EDTA). After the performance of the in vitro ubiquitination assay for gel-based analysis (see above), 500 nM of the diluted USP2 was added to the ubiquitination mix. The reaction mix was incubated for 1 h at 30°C, before terminating the reaction with LSB. SDS–PAGE analysis was performed as previously described in the in vitro ubiquitination assay for gel-based analysis.

### Ubi-real in vitro ubiquitination assay for fluorescent polarisation measurements

MARCH5-lipid samples were prepared as described above in vitro ubiquitination assay for gel-based analysis. A master solution resulting in final concentration of 40 mM Tris–HCl, pH 7.5, 25 mM NaCl, 10 mM MgCl$_2$, 0.6 mM DTT, 50 nM human E1 (#E-304; Boston Biochem), 1 $\mu$M E2 (UbcH5b, #E2-622; Boston Biochem), 23 $\mu$M Ub (#U6253; Sigma-Aldrich), and 100 nM F-Ub (#U-580; Boston Biochem) was mixed and aliquoted. 2.5 $\mu$M of MARCH5 incubated with lipids was added to the respective master solution or a master solution of MARCH5 without lipids as a + control. Concentrations were calculated so that desired concentrations would be achieved in a final volume of 20 $\mu$l. 18.4 $\mu$l of the master solution containing MARCH5 incubated with specific lipids were added to a Greiner 384-well, F-bottom, Hibase plate sample wells. The plate was sealed (Clear sealing tape; Molecular Dimensions). After baseline measurement, 1.6 $\mu$l ATP were added to each well to a final concentration of 4 $\mu$M, except for the–control, which included MARCH5 in the absence of lipids. The addition of ATP interrupted the measurement for around 2 min. The final detergent and lipid concentration in the master solution was 1 CMC (0.001%) 25 $\mu$M, respectively. The FP was monitored on a plate reader (Victor Nivo; Perkin Elmer) using a suitable setting for the fluorescein fluorophore with an excitation wavelength of 480, 500 nm dichroic mirror and emission wavelength of 530 nm. FP experiments were 2 h, and FP values were read every minute with a measuring time of 100 ms. In between measurements, the plate was shaken. The baseline was measured for 10 cycles (10 min). Each sample was prepared in four replicates, and the FP signal was averaged at each time point. FP signal was calculated by the Victor Nivo Software, and averages and standard deviations were calculated using GraphPad Prism. To note, fluorescent-labelled ubiquitin has a fluorophore permanently attached to its N-terminus that precludes methionine 1 (Met1)-linked ubiquitin chain formation. However, this ubiquitin linkage type is specific for the ubiquitin chain assembly complex LUBAC, which is the only known mammalian ubiquitin ligase that catalyses Met1-linked ubiquitin linkages (Hrdinka & Gyrd-Hansen, 2017).

### Chemical crosslinking in vitro

Prior experiment, MARCH5 was buffer exchanged using Zeba spin column (#89882; Thermo Fisher Scientific) to 50 mM Hepes, pH 7.7,

100 mM NaCl, 5% (vol/vol) glycerol, and 0.1 mM TCEP, 0.001% (wt/vol) LMNG. Bs3 crosslinking reagent (#21586; Thermo Fisher Scientific) was prepared in a 50 mM stock solution by dissolving 10 mg Bs3 in 350 $\mu$l of 25 mM potassium phosphate, pH 7.7. MARCH5 was incubated with CL for 1 h before the experiments in a 1:10 M ratio protein to lipid as described in the lipid preparation section. To 35 $\mu$l of 19.4 $\mu$M MARCH5, 0.35 $\mu$l Bs3 was added from the stock solution to a final concentration of 0.5 $\mu$M. At each time point (0, 15, 30, and 60 min), 8 $\mu$l was taken out of the MARCH5 Bs3 mix, and the reaction was quenched by adding 1 $\mu$l 3 M Tris–HCl, pH 7.7 followed by the addition of 4 $\mu$l LSB. Crosslinking was analysed by SDS–PAGE.

### Nano differential scanning fluorometry (nanoDSF)

Lipids were prepared as stated in the lipid preparation section. MARCH5 (final concentration of 0.5 mg/ml [16 $\mu$M]) was mixed with indicated lipids (stock concentration of 5 mM) in a 1:10 M ratio in a sample volume of 40 $\mu$l. The final detergent concentration was 5 CMC (0.005%). For the lipid titration experiments, 5 $\mu$l of desired stock concentration of lipids (see lipid preparation) to achieve the wanted molar ratio of protein to lipid (1:1, 1:5, 1:10, and 1:50) was added to MARCH5 (32 $\mu$M), having a final volume of 40 $\mu$l. After MARCH5 was diluted to a concentration of 0.5 mg/ml (16 $\mu$M) in buffer (50 mM Tris–HCl, pH 7.7; 100 mM NaCl; 5% glycerol; 0.1 mM TCEP; 0.001% [wt/vol] LMNG), the final detergent concentration was 12.5 CMC (0.0125%). Samples were incubated for 1 h at 4°C. Real-time simultaneous monitoring of the intrinsic tryptophan fluorescence at 330 and 350 nm was performed using a Prometheus Panta (Nanotemper) measuring a temperature range of 15°C–95°C with a temperature gradient of 1°C per minute. A minimum of triplicates was used for all experiments. Protein melting temperatures were determined as maxima of the first derivatives of the 350/330 nm ratio, analysed by Prometheus software.

## Supplementary Information

## Acknowledgements

We wish to thank M Cotta for her help in the initial phase of the ubiquitination assay establishment. The research leading to these results has received funding from the European Union Seventh Framework Programme (FP7-PEOPLE-2013-COFUND) under grant agreement no 609020 – Scientia Fellow (J Bauer). Figs 1A and C and 5 were created using https://biorender.com.

### Author Contributions

L Merklinger: conceptualization, resources, data curation, software, formal analysis, validation, investigation, visualization, and writing—original draft, review, and editing.
J Bauer: validation, methodology, and writing—review and editing.

PA Pedersen: resources, formal analysis, methodology, and writing—review and editing.

RB Damgaard: resources, software, formal analysis, supervision, investigation, methodology, and writing—review and editing.

JP Morth: conceptualization, resources, supervision, funding acquisition, validation, investigation, methodology, project administration, and writing—original draft, review, and editing.

## Conflict of Interest Statement

The authors declare that they have no conflict of interest.

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
