## [Reviewer comments · Life Science Alliance]

Life Science Alliance

Phospholipids alter activity and stability of mitochondrial membrane-bound ubiquitin ligase MARCH5

Lisa Merklinger, Johannes Bauer, Per Pedersen, Rune Damgaard, and Jens Preben Morth

DOI: <https://doi.org/10.26508/lsa.202101309>

Corresponding author(s): Jens Preben Morth, Technical University of Denmark

Review Timeline:	Submission Date:	2021-11-21
	Editorial Decision:	2021-12-28
	Revision Received:	2022-02-27
	Editorial Decision:	2022-03-17
	Revision Received:	2022-04-05
	Accepted:	2022-04-07

Scientific Editor: Novella Guidi

Transaction Report:

December 28, 2021

Re: Life Science Alliance manuscript #LSA-2021-01309-T

Prof. Jens Preben Morth
Technical University of Denmark
Department of Biotechnology and Biomedicine
Soltoft plads
Kgs Lyngby, Sjælland 2800
Denmark

Dear Dr. Morth,

Thank you for submitting your manuscript entitled "Phospholipids alter activity and stability of mitochondrial membrane-bound ubiquitin ligase MARCH5" to Life Science Alliance. The manuscript was assessed by expert reviewers, whose comments are appended to this letter. We invite you to submit a revised manuscript addressing the Reviewer comments.

Thank you for this interesting contribution to Life Science Alliance. We are looking forward to receiving your revised manuscript.

Sincerely,

B. MANUSCRIPT ORGANIZATION AND FORMATTING:

Reviewer #1 (Comments to the Authors (Required)):

Merklinger and colleagues investigate the regulation of MARCH5 by lipids, and report altered activity and stability specific to cardiolipin. The authors compare auto-ubiquitination by detergent solubilized MARCH5 over a range of lipid conditions and detergent concentrations, and also find cardiolipin binding induces a decrease in thermal stability, while phosphatidic acid increases thermal stability. A few questions/suggestions to help improve the manuscript:

1. The authors carefully perform titrations of various lipids. Have authors consider additional ways to distinguish the activity and stability of the MARCH5? Would mutants in the putative cardiolipin binding site or testing modified forms of cardiolipin (oxidized CL) provide additional support that the increased activity is indeed not a result of aggregation? Can higher-order species formation be reversed if a de-ubiquitinating enzyme is added?
2. The authors also mention MARCH6 regulation by cholesterol. What are the sequence differences between the different MARCH family members, and could this be informative in understanding lipid regulation?
3. Finally, the authors cite that there is an increased rate for auto-ubiquitination in the presence of cardiolipin. Could the authors comment on how this compares to typical rates for ubiquitination?

Overall, this is a well executed study. I recommend publication following minor revision.

Reviewer #2 (Comments to the Authors (Required)):

In this manuscript by Merklinger et al., the authors showed that mitochondrial membrane-bound ubiquitin ligase MARCH5 is controlled by phospholipid species. Cardiolipin and phosphatidic acid bound to purified MARCH5 and regulated the stabilization and the ubiquitination pattern. Some of the observations of the study are interesting, however some issues need to be addressed.

Major

- (1) Lipid binding assay using PVDF membrane was shown in Fig. 1B. Dose the detergent affect this result? Have the authors examined this experiment using a crude membrane fraction without detergent? CL signal was not clear. In Fig. C, is the degree of detergent replacement altered by the phospholipid species? Is this difference affect the result, Fig. 1D?
- (2) Authors showed the potential CL and PA binding sites. Mutagenetic assay such as replacement of some residues by alanine will enhance this manuscript in Figs 1-4.
- (3) In all assays, do fatty acid compositions of phospholipids affect each result? It is an important point.
- (4) In Fig. 5 and its discussion, authors suggested CL as regulator of MARCH5 upon externalization to the OMN. However, this manuscript did not show the results supporting the author's suggestion.

Minors

Authors should provide better explanation for Fig. 4 in the results and the legend.

Responses to reviewers' comments are in blue.

Line numbering refers to the revised version of the manuscript.

Reviewer #1 (Comments to the Authors (Required)):

Merklinger and colleagues investigate the regulation of MARCH5 by lipids, and report altered activity and stability specific to cardiolipin. The authors compare auto-ubiquitination by detergent solubilized MARCH5 over a range of lipid conditions and detergent concentrations, and also find cardiolipin binding induces a decrease in thermal stability, while phosphatidic acid increases thermal stability. A few questions/suggestions to help improve the manuscript:

We thank the reviewer for the appreciation of our manuscript.

1. The authors carefully perform titrations of various lipids. Have authors consider additional ways to distinguish the activity and stability of the MARCH5? Would mutants in the putative cardiolipin binding site or testing modified forms of cardiolipin (oxidized CL) provide additional support that the increased activity is indeed not a result of aggregation? Can higher-order species formation be reversed if a de-ubiquitinating enzyme is added?

We very much appreciate the comment and given the nature of our study, we did consider this particular issue. The binding of CL to MARCH5 and the induced destabilization might relate to higher conformational flexibility of MARCH5, which allows the auto-ubiquitination. It is difficult to assess if the binding of CL to MARCH5 is causing auto-ubiquitination or the decrease in thermal stability and therefore probably the higher conformational flexibility induced by CL, which might result in aggregation. The binding of CL or the possible aggregation is always directly or indirectly related to the enhanced auto-ubiquitination of MARCH5. Therefore, mutational studies would only confirm or reject the hypothesis of specific CL binding sites. However, would not give insight into what causes the auto-ubiquitination, the specific CL interaction or the conformational flexibility induced by CL. This would also be the case for oxidized CL. Whereat, currently there are no oxidized CL species commercially available.

Furthermore, mutational experiments always have to be accompanied with additional control experiments to confirm MARCH5 structural and functional integrity. For example, activity assay or additional stability experiments, since a single mutation could destabilize the protein to such a degree that it does not necessarily reflect the loss of functionality of the individual

amino acid. There is no experimental crystal structure available for MARCH5 or confident homology model, which might make it difficult to design the mutational residues. Additionally, there are inevitable limitations to the biophysical method one want to perform when working with membrane proteins, which also relates to MARCH5.

We included a cross-linking experiment of MARCH5 in presence of CL and in absence of lipids to see possible formation of higher oligomeric species in presence of CL (Fig S4, and added description in the figure caption in line 759-761, and main text line 191-194, materials and methods line 492-501). The SDS-PAGE gel showed that MARCH5 is probably present in a mix of dimer and monomer. A trend towards increased dimerization can be observed in the presence of CL, when compared to the control, but no aggregation.

The addition of a deubiquitinating enzyme reverses the ubiquitination reaction, supported by the reappearance of the mono-ubiquitination band and the reduction of the ubiquitin smear (comment added line number 162-164, included western blot visualizing MARCH5 and ubiquitin in Fig. S3). However, MARCH5 band is only marginally coming back. This is possibly because MARCH5 sticks to the loading pocket, pointing towards aggregation. However, since MARCH5 band was clearly visible at time point 0 in the ubiquitination assay in presence of all CL concentration allowing a nice separation in the SDS-PAGE gel, would be pointing towards that MARCH5 might have been aggregated upon the ubiquitination. This would explain the faint MARCH5 band in the de-ubiquitination assay.

2. The authors also mention MARCH6 regulation by cholesterol. What are the sequence differences between the different MARCH family members, and could this be informative in understanding lipid regulation?

The MARCH- family is defined and characterized by the cytosolic RING domain, via the arrangement of His and Cys residues, which coordinate the two zinc ions. The MARCH family contains a Cys and His on the fourth and fifth zinc coordinating residue (RING-CH), whereas the classic RING domain includes a His and a Cys (RING-HC), respectively (Dodd *et al*, 2004). The coordinating residues are highly conserved among the family. However, the MARCH family members highly vary in the number of transmembrane domains (some have no transmembrane domains) as well as their sequence. The transmembrane domain within the MARCH family is not conserved (Samji *et al*, 2014; Bauer *et al*, 2017). Therefore, we do not

think that we can conclude anything in regards to lipid regulation by looking at the sequence of the different transmembrane domains.

3. Finally, the authors cite that there is an increased rate for auto-ubiquitination in the presence of cardiolipin. Could the authors comment on how this compares to typical rates for ubiquitination?

This is an interesting question. However, we cannot conclude anything about the rate of auto-ubiquitination at the stage of experimental data we have and we do not discuss about rate of auto-ubiquitination but enhancement of auto-ubiquitination when compared to the control.

However, looking qualitatively at the time frame for ubiquitination, our observations are consistent with published results of ubiquitination assays from other RING E3s. The ubiquitination activity we observe *in vitro* for recombinant MARCH5 is comparable to those reported in the literature for other (soluble) RING E3 ligases where ubiquitination or auto-ubiquitination occurs at 10-60 min (Lips *et al*, 2020; Banka *et al*, 2015; Ranaweera & Yang, 2013; Branigan *et al*, 2020).

Overall, this is a well executed study. I recommend publication following minor revision.

We thank the referee for their thorough assessment of our work and their recommendation.

Reviewer #2 (Comments to the Authors (Required)):

In this manuscript by Merklinger et al., the authors showed that mitochondrial membrane-bound ubiquitin ligase MARCH5 is controlled by phospholipid species. Cardiolipin and phosphatidic acid bound to purified MARCH5 and regulated the stabilization and the ubiquitination pattern. Some of the observations of the study are interesting, however some issues need to be addressed.

We thank the reviewer for the appreciation of our study.

Major

(1) Lipid binding assay using PVDF membrane was shown in Fig. 1B. Dose the detergent affect this result? Have the authors examined this experiment using a crude membrane fraction without detergent? CL signal was not clear. In Fig. C, is the degree of detergent replacement altered by the phospholipid species? Is this difference affect the result, Fig. 1D?

We performed the lipid binding with two different detergent DDM and LMNG, which show the same results. Furthermore, we performed lipid-binding assays, including the lipids tested in the ubiquitination assay and stability measurements detecting with a MARCH5 antibody excluding possible interaction of GFP-His10 tag to interact with the lipids. We included an additional figure in the supplementary (Fig S1) and statement in line 107-108, and additional methods line 402-410.

We did not perform a lipid-binding assay using crude membrane fraction, because we would not expect MARCH5 binding to any of the lipids, since it is still incorporated in the natural lipid bilayer and would compete with the natural lipid carried over from the membrane.

The lipids were dissolved in the same detergent concentration to have same molar concentration in the lipid stock solution as well as the same molar ratio of protein to lipid in the experiment.

The detergent concentration does not have an effect on the thermal stability of MARCH5. This is shown by the lipid titration experiments, where the amount of detergent in the sample was 0.012% (w/v) LMNG (Table 2), which resulted in a melting temperature of 52.91 ± 0.65 °C. In contrast, the thermal stability measurement, where all the six different lipids were tested, the amount of detergent consisted of 0.005% (w/v) and resulted in a melting temperature of 52.86 ± 0.58 °C for the + control (Table1), which coincide with thermal stability measurement of the lipid titration experiments. This is also valid for the other lipids (see Table 1 and 2). Furthermore, there was no effect on the ubiquitination patters with variation in detergent concentration, showed by the ubiquitination assay of the lipid titration experiments, where the detergent concentration was 0.002% (w/v) (Fig. 3 and Fig S5), whereas the ubiquitination assay including all the lipids 0.001% (w/v) LMNG was present (Fig. 2)). Therefore, we can exclude that the detergent concentration has an influence on the thermal stability or stability.

Of note, the critical micelle concentration of the used detergent LMNG is 0.001% (1 CMC), which means at this concentration LMNG micelles are formed. Therefore, we always included detergent above this concentration. The detergent concentrations are stated in the respective material and method section.

(2) Authors showed the potential CL and PA binding sites. Mutagenetic assay such as replacement of some residues by alanine will enhance this manuscript in Figs 1-4.

We agree with the reviewer on this point; it would be interesting analyses to perform. However, these would be quite extensive experiments as many additional mutants will have to be purified and verified for stability with CD to shown that they otherwise behave similarly to the wild type enzyme. Therefore, given the scope of the study, we decided early in the process that we will focus on the wild type enzyme, mainly since we amount of protein and lipids we require for each experiment is quite large. See also the response to point 1 from Reviewer 1.

(3) In all assays, do fatty acid compositions of phospholipids affect each result? It is an important point.

This is a very interesting question, but it would be beyond the scope of this paper to analyse the impact of fatty acid composition in our assays. The experiments were performed using natural lipids occurring in vertebrate tissue (stated in material and methods). Therefore, the specific fatty acid composition was not considered, rather the overall lipid class. However, MARCH5 might interact only with specific lipid species, but this was not the aim of this study. We wanted to give a broad overview about possible effect of different lipid classes towards activity and stability of MARCH5. Testing different fatty acid compositions would exceed the scope of this initial study considering the enormous possibility of fatty acid combinations.

Furthermore, using natural lipids in this study had the advantage to cover large range of fatty acid compositions of the lipids occurring in the cell. This allows us to exclude specific binding for example of the abundant lipids PE and PC. We included a statement in line 131 and 132 that we use natural lipids, which is a mix of different lipid species. Furthermore, we replaced the word lipid species by the word lipid class to prevent confusion.

We are currently looking into specific fatty acid compositions of the lipids interacting with MARCH5 for a further study to evaluate the fatty acid dependency, which might be solely responsible for the effect on MARCH5. We show some initial results here (see figure below), where we tested CL of *E.coli* and the CL with the fatty acid chains 18:1. In the thermal stability measurement, we can see different effects on MARCH5. *E.coli* CL seems not to have an effect on the thermal stability compared to the control. However, CL 18:1 shows

minor stabilization. Initial ubiquitination assay reveals no significant change in presence of one of the CL species.

However, we do not know yet the cause of the difference, which we will investigate in the future, but CL is known to have specific fatty acid chains in different tissues, therefore might connect to different regulatory effects (Fajardo *et al*, 2017; Bradley *et al*, 2016).

Stability and activity measurement of MARCH5 in presence of CL species. (A) Differential scanning fluorimetry curve (left) and its derivative (right) of detergent-solubilised MARCH5 in presence of *E.coli* CL, 18:1 CL, Heart CL, or in absence of lipids (control) to assess thermal stability by tracking intrinsic fluorescence at 330 nm and 350 nm of mainly MARCH5 Tryptophan with increasing temperature. For clarity, one replicate of three independent experiments performed in triplicates is shown. (B) Melting temperatures of MARCH5 in presence of different CL species determined as the maxima of the first derivative of the 350nm/330nm ratio of the DSF curves shown in (A). Data are shown as mean values ± SD of three independent experiments, each performed in triplicates. (C-D) MARCH5 ubiquitination assay performed in presence of different CL species (18:1 and *E.coli* CL) in a

1:10 molar ratio protein to lipids and in absence of lipids (control) shown on a Coomassie stained SDS-PAGE gel (C) and western blot (D) targeted for MARCH5 (upper panel) and ubiquitin (lower panel). Ubiquitin reactions were terminated at different time points (0, 15, 30, 60 min).

(4) In Fig. 5 and its discussion, authors suggested CL as regulator of MARCH5 upon externalization to the OMN. However, this manuscript did not show the results supporting the author's suggestion.

We thank the reviewer for bringing our attention to this. The model depicted in Fig 5 is a speculative model that places our in vitro findings in a larger biological context and in relation to relevant literature. This have emphasized in the text, in line 293 and in the figure caption in line 732, 733, that this is a speculative model.

Minors

Authors should provide better explanation for Fig. 4 in the results and the legend.

We thank the reviewer for highlighting this. We have included the following explanation in the paragraph “MARCH5 stability is affected by the lipid phospholipid environment” in line 202-205 and lines 210, 214, 216 and 220 marked in red. Furthermore, we added an explanatory sentence in the caption in line 720 also marked in red.

References:

- Banka PA, Behera AP, Sarkar S & Datta AB (2015) RING E3-catalyzed E2 self-ubiquitination attenuates the activity of Ube2E ubiquitin-conjugating enzymes. *J Mol Biol* 427: 2290–2304
- Bauer J, Bakke O & Morth JP (2017) Overview of the membrane-associated RING-CH (MARCH) E3 ligase family. *N Biotechnol* 38: 7–15
- Bradley RM, Stark KD & Duncan RE (2016) Influence of tissue, diet, and enzymatic remodeling on cardiolipin fatty acyl profile. *Mol Nutr Food Res* 60: 1804–1818
- Branigan E, Carlos Penedo J & Hay RT (2020) Ubiquitin transfer by a RING E3 ligase occurs from a closed E2~ubiquitin conformation. *Nat Commun* 11: 1–11
- Dodd RB, Allen MD, Brown SE, Sanderson CH, Duncan LM, Lehner PJ, Bycroft M & Read RJ (2004) Solution structure of the Kaposi's sarcoma-associated herpesvirus K3 N-terminal domain reveals a novel E2-binding C4HC3-type RING domain. *J Biol Chem* 279: 53840–53847
- Fajardo VA, Mikhaeil JS, Leveille CF, Saint C & LeBlanc PJ (2017) Cardiolipin content, linoleic acid composition, and tafazzin expression in response to skeletal muscle overload and unload stimuli. *Sci Rep* 7: 1–9
- Lips C, Ritterhoff T, Weber A, Janowska MK, Mustroph M, Sommer T & Klevit RE (2020) Who with whom: functional coordination of E2 enzymes by RING E3 ligases during poly-ubiquitylation. *EMBO J* 39: 1–20
- Ranaweera RS & Yang X (2013) Auto-ubiquitination of Mdm2 enhances its substrate ubiquitin ligase activity. *J Biol Chem* 288: 18939–18946
- Samji T, Hong S & Means RE (2014) The Membrane Associated RING-CH Proteins: A Family of E3 Ligases with Diverse Roles through the Cell. *Int Sch Res Not* 2014: 1–23

March 17, 2022

RE: Life Science Alliance Manuscript #LSA-2021-01309-TR

Prof. Jens Preben Morth
Technical University of Denmark
Department of Biotechnology and Biomedicine
Soltoft plads
Kgs Lyngby, Sjælland 2800
Denmark

Dear Dr. Morth,

Thank you for submitting your revised manuscript entitled "Phospholipids alter activity and stability of mitochondrial membrane-bound ubiquitin ligase MARCH5". We would be happy to publish your paper in Life Science Alliance pending final revisions necessary to meet our formatting guidelines.

- please upload your main and supplementary figures as single files
- please add Keywords and Category for your manuscript in our system
- please note that the titles in the system and manuscript file must match
- please add callouts for Figures S1A-B; S3A-B; S4A-C and S5A, B, D-F to your main manuscript text;
- there is a splice between the second and third set of blots in figure 2B, third panel. Also there are a few flaws in the blots in Figure 3B (e.g. white stripes). Same for S5B. Please provide source data files for figure 2B, 3B and S5B.

A. FINAL FILES:

B. MANUSCRIPT ORGANIZATION AND FORMATTING:

Sincerely,

Reviewer #1 (Comments to the Authors (Required)):

The authors have addressed all comments from this Reviewer.

It is a nice piece of work. I recommend publication without delay.

Reviewer #2 (Comments to the Authors (Required)):

In this manuscript by Merklinger et al., the authors showed that phospholipid species affect mitochondrial membrane-bound ubiquitin ligase MARCH5. Purified MARCH5 bound to cardiolipin and phosphatidic acid. Additionally, phospholipids regulated the stabilization and the ubiquitination pattern. This manuscript suggest that phospholipids can regulate the dynamics and turnover of mitochondria.

The authors have satisfyingly answered all my questions.

April 7, 2022

RE: Life Science Alliance Manuscript #LSA-2021-01309-TRR

Prof. Jens Preben Morth
Technical University of Denmark
Department of Biotechnology and Biomedicine
Soltoft plads
Kgs Lyngby, Sjælland 2800
Denmark

Dear Dr. Morth,

Thank you for submitting your Research Article entitled "Phospholipids alter activity and stability of mitochondrial membrane-bound ubiquitin ligase MARCH5". It is a pleasure to let you know that your manuscript is now accepted for publication in Life Science Alliance. Congratulations on this interesting work.

DISTRIBUTION OF MATERIALS:

Again, congratulations on a very nice paper. I hope you found the review process to be constructive and are pleased with how the manuscript was handled editorially. We look forward to future exciting submissions from your lab.

Sincerely,
